# Enhancement of High-Density Lipoprotein (HDL) Quantity and Quality by Regular and Habitual Exercise in Middle-Aged Women with Improvements in Lipid and Apolipoprotein Profiles: Larger Particle Size and Higher Antioxidant Ability of HDL

**DOI:** 10.3390/ijms24021151

**Published:** 2023-01-06

**Authors:** Kyung-Hyun Cho, Hyo-Seon Nam, Dae-Jin Kang, Seonggeun Zee, Min-Hee Park

**Affiliations:** 1Raydel Research Institute, Medical Innovation Complex, Daegu 41061, Republic of Korea; 2LipoLab, Yeungnam University, Gyeongsan 38541, Republic of Korea

**Keywords:** high-density lipoproteins, apolipoprotein A-I, exercise, paraoxonase, low-density lipoproteins

## Abstract

Regular exercise, especially aerobic exercise, is beneficial for increasing serum high-density lipoprotein-cholesterol (HDL-C) levels in the general population. In addition to the HDL-C quantity, exercise enhances HDL functionality, antioxidants, and cholesterol efflux. On the other hand, the optimal intensity and frequency of exercise to increase HDL quantity and enhance HDL quality in middle-aged women need to be determined. The current study was designed to compare the changes in HDL quantity and quality among middle-aged women depending on exercise intensity, frequency, and duration; participants were divided into a sedentary group (group 1), a middle-intensity group (group 2), and a high-intensity group (group 3). There were no differences in anthropometric parameters among the groups, including blood pressure, muscle mass, and handgrip strength. Although there was no difference in serum total cholesterol (TC) among the groups, the serum HDL-C and apolipoprotein (apo)A-I levels remarkably increased to 17% and 12%, respectively, in group 3. Serum low-density lipoprotein-cholesterol (LDL-C), glucose, triglyceride, and the apo-B/apoA-I ratio were remarkably decreased in the exercise groups depending on the exercise intensity; group 3 showed 13%, 10%, and 45% lower LDL-C, glucose, and triglyceride (TG), respectively, than group 1. The hepatic and muscle damage parameter, aspartate aminotransferase (AST), was significantly decreased in the exercise groups, but high-sensitivity C-reactive protein (CRP), alanine aminotransferase (ALT), and γ-glutamyl transferase (γ-GTP) were similar in the three groups. In LDL, the particle size was increased 1.5-fold (*p* < 0.001), and the oxidation extent was decreased by 40% with a 23% lower TG content in group 3 than in group 1. In the exercise groups (groups 2 and 3), LDL showed the slowest electromobility with a distinct band intensity compared to the sedentary group (group 1). In HDL_2_, the particle size was 2.1-fold increased (*p* < 0.001) in the exercise group (group 3) with a 1.5-fold increase in TC content compared to that in group 1, as well as significantly enhanced antioxidant abilities, paraoxonase (PON) activity, and ferric ion reduction ability (FRA). In HDL_3_, the particle size was increased 1.2-fold with a 45% reduction in TG in group 3 compared to group 1. With increasing exercise intensity, apoA-I expression was increased in HDL_2_ and HDL_3_, and PON activity and FRA were enhanced (*p* < 0.001). In conclusion, regular exercise in middle-aged women is associated with the elevation of serum HDL-C and apoA-I with the enhancement of HDL quality and functionality and an increase in the TC content, particle size, and antioxidant abilities. With the reduction in TG and oxidized products in LDL and HDL, lipoproteins could have more anti-atherogenic properties through regular exercise in an intensity-dependent manner.

## 1. Introduction

Sedentary lifestyles are a major risk factor for metabolic syndrome and cardiovascular diseases [1,2]. A sedentary lifestyle is frequently associated with low HDL-C and high triglyceride (TG), insulin resistance, and abdominal obesity [3,4]. Exercise is associated with a reduced risk of cardiovascular disease and reduced overall mortality in a dose-dependent manner [5,6]. Regular exercise is beneficial for increasing HDL-C with enhanced functionality, such as antioxidant activity, anti-inflammatory activity, and cholesterol efflux activity [7,8].

HDL-C quantity and quality can be impaired by sedentary lifestyles and poor dietary habits [9], alcohol consumption [10], and smoking [11]. The production of dysfunctional HDL is intimately associated with a high incidence of metabolic syndrome because impaired qualities of HDL, such as decreased oxidation and glycation properties, are frequently found in patients with obesity, diabetes, and hypertension [12]. Therefore, it is widely accepted that maintaining HDL to retain its healthy and protective effects is associated with the suppression of hypertension, diabetes, and dementia [13]. In general, the quantity of HDL-C gradually decreases with aging [14], which impairs HDL quality [15].

HDL-C quantity and quality can also be enhanced through lifestyle and nonpharmacological interventions, such as exercise [16] and nutritional supplementation [17]. As an enhancement strategy, many human studies have shown that nutrient supplementation, such as policosanol, could enhance HDL-C quantity, quality, and functionality in healthy subjects [18,19] with prehypertension [20,21]. Enhanced HDL functionality was closely associated with increased apoA-I and paraoxonase activity in HDL [16,19,20]. Aerobic sports athletes, such as runners and wrestlers, have the highest HDL-C and the largest particle size with the highest content of apoA-I and paraoxonase (PON) activity in HDL among national representatives of the Beijing Olympics [16]. These results suggest that regular exercise could enhance HDL functionality and increase HDL-C quantity regardless of the exercise type.

Many reports have shown that the pro-atherogenic lipid profile was improved to an anti-atherogenic profile through exercise, as illustrated by decreases in LDL-C and TG and an increase in HDL-C [22]. On the other hand, there are no reports comparing HDL quality and functionality between sedentary and exercise groups in nonsmoking middle-aged women. This study was designed to compare changes in HDL and LDL properties, such as particle shape and size, oxidation and glycation extent, and lipid and apolipoprotein compositions, depending on the exercise intensity in healthy and normolipidemic middle-aged women who carried out regular exercise for at least one year prior to the study.

## 2. Results

### 2.1. Anthropometric Profiles

As shown in Table 1, all participants (n = 57) were divided into three groups depending on the metabolic equivalent score based on exercise time and intensity: sedentary (group 1, MET 2.48 ± 0.03), low-exercise (group 2, moderate intensity for 60 min/week, MET 4.55 ± 0.27), and high-exercise (group 3, high intensity for 90 min/week, 9.13 ± 0.35) groups. The three groups comprised females of similar age (around 50 years of age; 35–63 years old) and similar occupations, such as office and indoor work. Regardless of the exercise time and intensity, anthropometric data, such as BMI, blood pressure, muscle mass, fat mass, and handgrip strength (HS), were similar among the groups. All participants showed normal and similar ranges of anthropometric profiles for middle-aged women and were healthy without diagnosed diseases (Table 1).

**Table 1 ijms-24-01151-t001:** Anthropometric profiles of women groups depending on exercise intensity (Me (25%; 75%)) *.

	Group 1 n = 20	Group 2 n = 16	Group 3 n = 21	*p*	1 vs. 2	1 vs. 3	2 vs. 3
Moderate-intensity exercise/week (times)	0.0 (0.0; 0.0)	1.5 (1.5; 3.5)	3.5 (1.5, 5.5)	<0.001	0.000	0.000	0.671
High-intensity exercise/week (times)	0.0 (0.0; 0.0)	0.0 (0.0; 0.0)	1.5 (1.5, 3.5)	<0.001	1.000	<0.001	<0.001
Total number of exercises/week (times)	0.0 (0.0; 0.0)	1.5 (1.5; 3.5)	5.0 (3.0, 7.0)	<0.001	<0.001	<0.001	0.009
Total time of exercise/week (minutes)	0 (0; 0)	60 (60; 150)	90 (75; 150)	<0.001	<0.001	<0.001	0.357
MET equivalents of common physical activities (score)	2.5 (2.5; 2.5)	4.8 (3.3; 5.5)	9.5 (8.0; 10.0)	0.000	0.000	0.000	0.000
Alcohol intake/month (g)	63 (5; 210)	21 (0; 94.5)	21 (0, 137)	0.458	0.801	0.901	1.000
Age (year), (min., max.)	53.5 (50.0; 56.0) (41, 58)	45.0 (40.8; 54.8) (35, 63)	52.0 (43.0, 55.5) (36, 62)	0.132	0.171	0.301	0.924
Height (cm)	160.0 (157; 163.8)	161.0 (158.3; 165.0)	159.0 (156.0, 164.5)	0.804	1.000	1.000	1.000
Weight (kg)	55.7 (50.9; 59.9)	56.9 (53.3; 66.8)	54.5 (51.2; 61.1)	0.116	0.242	1.000	0.169
BMI (kg/m^2^), (min., max.)	21.4 (20.1; 22.8) (19.5, 26.1)	23.0 (20.4; 24.2) (18.4, 30.9)	21.4 (20.2, 23.0) (17.0, 26.8)	0.446	0.797	1.000	0.773
Heart rate (BPM)	70.0 (66.8; 75.8)	73.0 (66.0; 84.8)	75.0 (72.0, 81.0)	0.142	0.549	0.167	1.000
SBP (mmHg)	120.5 (110.0; 129.8)	123.0 (108.5; 127.5)	120.0 (117.0, 124.0)	0.930	1.000	1.000	1.000
DBP (mmHg)	75.5 (63.3; 87.8)	68.0 (62.8; 77.0)	69.0 (65.5; 78.0)	0.207	0.262	0.673	1.000
Muscle mass (kg)	37.5 (35.8; 38.8)	39.7 (35.4; 42.8)	38.1 (34.5; 40.1)	0.172	0.248	1.000	0.357
Fat mass_subcutaneous (kg)	13.6 (12.2; 15.9)	14.4 (12.7; 18.1)	13.7 (11.0; 16.4)	0.240	0.860	1.000	0.282
Fat mass_visceral (kg)	1.5 (1.2; 2.0)	1.8 (1.2; 2.4)	1.5 (1.1; 2.0)	0.406	1.000	1.000	0.542
HS (kg), (min., max.)	26.5 (23.0; 28.0) (16.0, 30.0)	25.0 (23.0; 27.8) (22.0, 37.0)	28.0 (25.5; 30.0) (13.0, 32.0)	0.158	1.000	0.275	0.334
Body water (kg)	29.2 (27.4; 30.3)	30.9 (27.6; 33.5)	29.9 (31.5)	0.087	0.103	1.000	0.263

BMI, body mass index; BPM, beat per minute, DBP, diastolic blood pressure; HS, handgrip strength; SBP, systolic blood pressure. * Data are presented as median (25th; 75th percentiles). The metabolic equivalent (MET) score was calculated based on the survey results of the participants and can be classified as previously described [23]: Light < 3.0 METs; Moderate 3.0~6.0 METs; Vigorous > 6.0 METs.

### 2.2. Blood Parameters

In the blood lipid profiles, all groups showed similar total cholesterol (TC) and low-density lipoprotein-cholesterol (LDL-C) levels of 202–222 mg/dL and 134–153 mg/dL, respectively (Table 2), with 66–68% of LDL-C in TC. However, HDL-C increased significantly with exercise intensity (*p* = 0.030); group 3 showed 15% higher HDL-C than group 1 (*p* = 0.040), while groups 1 and 2 did not show a difference. In the same context, the % of HDL-C in TC was more elevated in the exercise groups than the nonexercise group (*p* = 0.002) and showed the highest ratio of HDL-C in TC, around 28% (*p* < 0.001 vs. group 1), while groups 1 and 2 showed 22% and 26% of the %HDL-C/TC level in group 3. The serum TG levels remarkably decreased in the exercise groups compared to the nonexercise group (*p* < 0.001); group 3 showed a 45% lower TG level than group 1 (*p* < 0.001), while group 2 showed 23% lower TG than group 1 with no significance (*p* = 0.165). Concomitantly, the TG/HDL-C ratio and LDL-C/HDL-C ratio were decreased in the exercise groups; group 3 showed significantly lower levels than group 1 (*p* = 0.001).

Among apolipoprotein levels, apoA-I was increased and apo-B was decreased in the exercise groups, but the difference was not significant. On the other hand, the apo-B/apoA-I ratio was significantly lower in the exercise groups than in the nonexercise group (*p* = 0.014); group 3 showed a 23% lower apo-B/apoA-I ratio (0.52 ± 0.02) than group 1 (0.67 ± 0.04, *p* = 0.009). The serum glucose level was also lower in the exercise groups than in the nonexercise group (*p* = 0.003); group 3 showed a 10% lower serum glucose level than group 1 (*p* = 0.003). These results suggest that high-intensity exercise was associated with an increase in apoA-I, a decrease in apo-B, and a decrease in glucose in serum.

For hepatic functions, the determination of hepatic enzymes and acute inflammation parameters showed that the exercise groups had a significant reduction in aspartate aminotransferase (AST, *p* = 0.002) (Table 2). In contrast, none of the groups showed differences in alanine aminotransferase (ALT, *p* = 0.088), gamma-glutamyl transferase (γ-GTP, *p* = 0.064), or high-sensitivity C-reactive protein (hs-CRP, *p* = 0.896). Groups 3 and 2 showed 25% (*p* = 0.003) and 21% (*p* = 0.028) lower AST levels, respectively, than group 1. Group 3 showed 23% lower ALT and 35% lower γ-GTP levels than group 1, although there was no significance among any of the groups. These results suggest that exercise is specifically associated with a reduction in AST, but not ALT, because AST is more specifically influenced by muscle function and its damage than ALT, which is more specific to hepatic damage. The other enzymes, ALT and γ-GTP, are more related to hepatic-specific inflammation and alcoholic damage, respectively.

### 2.3. LDL Compositions and Particle Analysis

As shown in Table 3, the determination of the glycation extent using fluorescence spectroscopy showed that group 3 showed 7% lower fluorescence intensity (FI) than group 1, while group 2 showed a similar level to group 1. This result suggests that only high-intensity exercise, not low-intensity exercise, lowered the glycation of LDL compared to the sedentary group. The quantification of the TC content showed that group 1 had the lowest TC content in LDL, while groups 2 and 3 had 10% and 13%, respectively, higher TC content than group 1, but no significance was detected in the group comparison. The quantification of TG in LDL showed that the exercise groups had lower TG levels than the sedentary group (*p* = 0.023); group 3 had 23% lower TG content in LDL than group 1 (*p* = 0.041). These results suggest that high-intensity exercise is associated with an increase in TC and a decrease in TG in LDL.

### 2.4. Electron Microscopic Analysis of LDL

Transmission electron microscopy (TEM) showed that LDL from group 3 had the largest diameter of approximately 27.9 ± 0.5 nm (Table 3) and the largest size of 711 ± 13 nm^2^ (Figure 1A) with the most distinct shape (Figure 1B), while LDL from group 1 showed the smallest diameter, 23.9 ± 0.4 nm, and the smallest size, around 462 ± 8 nm^2^ (*p* < 0.001), with a more aggregated and ambiguous particle shape. Groups 2 and 3 showed larger diameters, 24.6 ± 0.3 nm and 27.9 ± 0.5 nm, respectively, and 1.2-fold (*p* < 0.001) and 1.5-fold (*p* < 0.001) larger LDL particle sizes, respectively, than group 1, indicating that higher-intensity exercise caused a larger LDL size with a more distinct morphology. Cupric-ion-mediated oxidized LDL (oxLDL) showed the most severe aggregated particle shape and unclear morphology with a diminished particle number (Figure 1). oxLDL showed the most aggregated and obscure image with the smallest particle size, around 402 ± 13 nm^2^. These results suggest that higher-intensity exercise was associated with a larger LDL particle size with a larger increase in cholesterol and a larger decrease in the TG content, glycation extent, and oxidation extent.

### 2.5. Electromobility of LDL

As shown in Figure 2A, LDL electrophoresis on agarose gel showed that the high-exercise group (group 3) had the strongest and distinct band intensity without aggregation in the loading position (lane 3). In contrast, group 1 (no exercise) showed the fastest electromobility and the highest smeared band intensity (lane 1). The low-exercise group (lane 2) showed a lower smeared band intensity and slower electromobility than the sedentary group (lane 1). LDL oxidized by the cupric ion treatment (lane 4) showed the fastest electromobility with the largest smeared band pattern and aggregation at the loading position, as indicated by the red arrowhead. Interestingly, the LDL band intensity was stronger and more distinct depending on the exercise intensity and the increase in MET. More oxidized LDL was prone to aggregation in the loading position due to increased apo-B fragmentation in LDL with the fastest electromobility, as indicated by the blue arrow and red arrowhead in Figure 1A.

The quantification of oxidized species in LDL showed that group 1 had the highest oxidized species level, around 1.1 ± 0.1 nM malondialdehyde (MDA), while groups 2 and 3 showed a lower MDA level, around 0.8 ± 0.1 nM and 0.6 ± 0.1 nM MDA, respectively. These results suggest that LDL oxidation was attenuated by exercise in an intensity-dependent manner; LDL from groups 2 and 3 showed a 28% and 45%, respectively, lower oxidation extent than group 1.

### 2.6. HDL_2_ Compositions and Particle Morphology

In HDL_2_, as shown in Table 3, the TC content was increased (*p* = 0.031) depending on the exercise intensity; group 3 showed the highest TC (*p* = 0.042) and the lowest TG content in HDL_2_. Groups 2 and 3 showed 1.4-fold and 1.7-fold higher TC contents than group 1, respectively, while group 3 showed 1.2-fold higher TC than group 2. Although no significance was detected, the TG content was lower in the exercise groups than in the sedentary group; group 3 showed 29% lower TG than group 1. There was no difference in the TG content in HDL_2_ between groups 1 and 2, suggesting that low-intensity exercise did not affect the change in TG content in HDL_2_.

TEM image analysis revealed that the HDL_2_ particle diameter in group 3 was 1.4 times (*p* < 0.001) and 1.2 times larger than those in groups 1 and 2, respectively, as shown in Figure 3. The particle size showed an increasing tendency in the exercise groups depending on intensity (*p* < 0.001); group 3 showed the largest size (234 ± 5 nm^2^), which was twice that of group 1 (114 ± 4 nm^2^, *p* < 0.001), with a distinct shape and clear morphology. Group 2 (162 ± 4 nm^2^) showed a 1.4 times larger HDL_2_ level than group 1, *p* < 0.001, while HDL_2_ glycated by fructose treatment (final 250 mM) for 24 h showed the smallest particle size with the most ambiguous morphology.

Fluorescence spectroscopy showed that the glycation extent was lower in the exercise groups; groups 2 and 3 showed 8% and 10% lower glycation than group 1. The extent of oxidation was lower in the exercise groups than in the sedentary groups, up to 23% lower in group 2 than in group 1, but the difference was not significant.

### 2.7. HDL_3_ Compositions and Particle Analysis

Although there was no difference in TC content in HDL_3_ among any of the groups, the exercise groups showed a lower TG content (*p* = 0.022) than the sedentary group; groups 2 and 3 showed up to 34% (*p* = 0.051) and 45% (*p* = 0.017) lower TG levels, respectively, than group 1. These results suggest that exercise helps reduce the TG content in HDL_3_, even though there was no accumulation of TC in HDL_3_. On the other hand, there was no difference in the oxidation extent or glycation extent among the groups.

The HDL_3_ particle diameter increased significantly in an exercise-intensity-dependent manner (*p* < 0.001); group 3 showed a 33% and 18% larger diameter than groups 1 and 2, respectively. The HDL size also increased with exercise intensity, as shown in Figure 4; group 3 showed the largest particle size, around 110 ± 13 nm^2^, whereas group 1 showed the smallest HDL_3_ size, around 92 ± 2 nm^2^. HDL_3_ in groups 2 and 3 showed an 11% (*p* = 0.004) and 20% (*p <* 0.001), respectively, bigger size than that in group 1, while group 3 showed an 8% bigger (*p* = 0.041) particle size than that in group 2. Glycated HDL_3_, which was fructose-treated (final 250 mM), showed the most significant decrease, around 58 ± 4 nm^2^. The extent of oxidation and glycation in HDL_3_ were similar between the groups, as shown in Table 3.

### 2.8. Electrophoretic Patterns of HDL_2_ and HDL_3_

As shown in Figure 5A, SDS-PAGE of HDL_2_ (2 mg/mL) and HDL_3_ (2 mg/mL) revealed that the exercise groups showed higher apoA-I expression, which increased in an intensity-dependent manner; groups 2 and 3 showed 1.3-fold and 1.6-fold higher band intensities of apoA-I, respectively, than group 1. Interestingly, group 3 showed the lowest band intensity of apoA-II in HDL_2_, even though group 3 showed the highest apoA-I content. Agarose electrophoresis revealed that the two distinct bands of HDL_2_ were the strongest in group 3, as shown in Figure 5B, while group 1 showed the two weakest bands with slower electromobility.

### 2.9. Correlation Analysis

As shown in Table 4, Spearman correlation analysis showed that higher-intensity exercise is positively and significantly associated with the serum HDL-C level (r = 0.365, *p* = 0.005), %HDL-C/TC (r = 0.484, *p* < 0.001), and the apoA-I level (r = 0.354, *p* = 0.007). On the other hand, higher-intensity exercise is negatively associated with serum apo-B (r = −0.264, *p* = 0.047), the apo-B/apoA-I ratio (r = −0.373, *p* = 0.004), and the serum LDL-C/HDL-C ratio (r = −0.437, *p* = 0.001). These results suggest that higher-intensity exercise is positively correlated with HDL-related parameters and negatively correlated with LDL-related parameters. Hepatic parameters, AST, ALT, and γ-GTP, were negatively correlated with exercise intensity: AST (r = −0.555, *p* < 0.001), ALT (r = −0.351, *p* = 0.007), and γ-GTP (r = −0.365, *p* = 0.005). On the other hand, hsCRP was not correlated with exercise intensity, suggesting that acute infection or inflammation was not correlated with exercise in healthy subjects.

For the lipoprotein level, the MDA content in LDL was negatively (r = −0.238, *p* < 0.001) correlated and the LDL size (r = 0.271, *p* < 0.001) was positively correlated with exercise in an intensity-dependent manner. In HDL_2_, the TC content, particle size, and PON activity were positively correlated with exercise intensity, even though FRA did not show a correlation with exercise intensity. In HDL_3_, FRA, PON activity, and particle size were positively correlated with exercise intensity, while the TC, TG, and MDA contents did not correlate with exercise intensity.

### 2.10. Antioxidant Activity in HDL_2_ and HDL_3_

Among the three groups, at the same protein concentration (2 mg/mL), HDL_3_ showed higher paraoxonase (PON) and ferric ion reduction ability (FRA) than HDL_2_, as shown in Figure 6, suggesting that HDL_3_ displayed a higher antioxidant capacity than HDL_2_. As shown in Figure 6A, the HDL-associated paraoxonase (PON) assay revealed that group 3 showed the highest PON activity in both HDL_2_ and HDL_3_ with 20.5 ± 1.3 μU/L/min and 41.8 ± 2.3 μU/L/min, respectively. On the other hand, group 1 showed the lowest PON activity in both HDL_2_ and HDL_3_ with 15.1 ± 0.7 μU/L/min and 31.6 ±0.8 μU/L/min, respectively, whereas group 2 showed higher PON activity in both HDL_2_ and HDL_3_ with 17.1 ± 1.3 μU/L/min and 34.9 ± 1.7 μU/L/min, respectively. These results suggest that the PON activity is significantly and proportionally associated with exercise in an intensity-dependent manner; groups 2 and 3 showed up to 12% and 35% higher HDL_3_-PON activity, respectively, than group 1.

The FRA of HDL_2_ and HDL_3_ was higher in both exercise groups and increased in a dose-dependent manner. In HDL_2_, group 3 showed the highest FRA of around 89 ± 5 μM, while group 1 showed the lowest FRA of around 69 ± 4 μM, although there were no significant differences in the group comparison. In HDL_3_, FRA increased significantly with exercise intensity (*p* < 0.001); groups 2 and 3 showed 14% (*p* = 0.048) and 40% (*p* < 0.001) higher FRA, respectively, than group 1. Interestingly, group 3 showed 22% higher FRA than group 2 (*p* = 0.012), suggesting that the enhancement of FRA is strongly dependent on exercise intensity.

Spearman correlation analysis showed that the PON activity in HDL_2_ (r = 0.714, *p* < 0.001) and HDL_3_ (r = 0.811, *p* < 0.001) was positively associated with exercise intensity. FRA in HDL_2_ (r = 0.578, *p* = 0.004) and HDL_3_ (r = 0.686, *p* < 0.001) was also positively correlated with exercise intensity.

## 3. Discussion

Regular exercise is associated with the elevation of serum HDL-C [24], apoA-I content [25], and cholesterol efflux activity with an increase in particle size [26]. A meta-analysis of 10 interventions showed that the number of large HDL particles increased despite the differences in exercise programs [27]. Although it has been widely accepted that HDL quality, including HDL particle size, is increased by exercise, there has been insufficient information on the particle shape, composition, and extent of oxidation and glycation. Moreover, many studies have focused on the short-term physiological effect of exercise, usually 8–24 weeks, which was intentionally forced with the randomization of volunteers [28]. In addition, there have been no studies that compare the effect of regular exercise, for at least 1 year or more, on HDL-C quantity and quality between a sedentary group and an exercise group in the general population. The current study aimed to find differences in HDL quality and functionality specifically in middle-aged women depending on whether they were sedentary or exercised.

The current study showed a difference in physiological parameters, including lipids, lipoproteins, apolipoproteins, hepatic functions, and inflammatory profiles, among healthy middle-aged women who carried out habitual and regular exercise. They were divided into sedentary (group 1), low-intensity exercise (group 2), and high-intensity exercise (group 3) based on a self-reported questionnaire. The exercise groups, particularly group 3, exhibited a significant elevation of serum HDL-C and %HDL-C/TC, with a significant reduction in TG, TG/HDL-C, LDL-C/HDL-C, and apo-B/apoA-I. In addition to improving the lipid profile, lipoprotein profile, and apolipoprotein profile, serum AST was significantly lower in groups 2 (*p* = 0.028 versus group 1) and 3 (*p* = 0.003 versus group 1) despite similar serum ALT and γ-GTP. Correlation analysis also showed that serum AST showed the highest negative association (r = −0.555, *p* < 0.001) with exercise intensity, while serum ALT (r = −0.351, *p* = 0.007) and γ-GTP (r = −0.365, *p* = 0.005) showed a less negative association. Similarly, exercise decreased both AST and ALT levels in patients with nonalcoholic fatty liver diseases [29]. From 16 clinical trials, the average reduction in the AST level was −4.93 U/L (95% CI, −7.94 to −1.91), which shows good agreement with the current findings. AST is more specific to muscle damage than ALT, which is produced primarily in the cytoplasm of hepatocytes and is more specific to liver damage [30]. AST is mainly produced in the mitochondria of the muscle, heart, kidney, red blood cells, brain, and small bowel, whereas ALT is present in the liver, muscle, and kidney [29]. The exercise groups showed lower AST (Table 2) with significant negative correlations (Table 4) regardless of age because AST is more specific to muscle damage. To the best of the authors’ knowledge, this is the first report to show that habitual regular exercise for at least one year, not through a forced exercise intervention, could reduce serum AST in middle-aged women.

In group 3, the serum apoA-I level was elevated, and the apoA-I band in HDL_2_ and HDL_3_ was increased, while apoA-II in HDL_2_ was decreased. These reciprocal changes in the expressional levels of apoA-I and apoA-II are similar to a previous report showing that Olympic athletes showed the elevation of apoA-I in HDL_2_ and HDL_3_, while their apoA-II was decreased, particularly in HDL_2_. [16] ApoA-I and apoA-II comprised around 70% and 20% of the total HDL protein content, respectively. Although apoA-II is the second most abundant protein constituent of HDL, apoA-II is still an enigmatic apolipoprotein. HDL containing apoA-I alone (LpA-I) and HDL containing apoA-I and apoA-II (LpA-I:A-II) were distinctly different, both structurally and metabolically [31]; LpA-I:A-II is more pro-atherogenic [32]. LpA-I is more cardioprotective in patients with coronary artery disease than LpA-I-I:A-II. Bioinformatic analysis showed that Lp-A-I is more metabolically active in facilitating the increase in HDL particle size and number [32]. LpA-I:A-II is less metabolically active and usually has a smaller particle size than LpA-I [33]. Overall, the increase in apoA-I and decrease in apoA-II in HDL_2_ (Table 2 and Figure 5A) in group 3 were linked to an increase in HDL_2_ particle size (Figure 3).

There have been conflicting data concerning the effect of exercise on improving HDL-C quantity, depending on the type of sport [34]; aerobic exercise effectively improves HDL-C, but resistance exercise does not. Aerobic exercise performed for 12–24 weeks can increase HDL-C more efficiently by approximately 3.8–15.4 mg/dL from the initial level [34]. On the other hand, these data were obtained from short-term or long-term exercise interventions that were carried out for training purposes or as part of mandatory programs. Furthermore, an increase in HDL-C quantity was not accompanied by the enhancement of HDL quality, such as increases in particle size and antioxidant ability. Healthy young men who completed a 12-week moderate-intensity exercise program as military soldiers [35] showed an elevation of HDL-C and apoA-I but without an increase in HDL particle size, even though their cholesterol efflux was enhanced. On the other hand, the anti-inflammatory properties of HDL were enhanced in these soldiers, but the antioxidant ability of HDL was not determined. Interestingly, the HDL particle size was increased with enhanced antioxidant activity in middle-aged women who had performed habitual exercise for at least one year with high intensity. The enhancement of HDL quality was accompanied by increases in the quantities of serum HDL-C and apoA-I that depended on the exercise intensity, suggesting that long-term exercise could improve the quantity and functionality of HDL. These findings can be extrapolated to women with higher BMI, since all groups had overweight subjects, especially group 2, which showed a BMI range of 18.9–30.9.

A limitation of this study was that the data on exercise frequency and intensity were obtained from self-reported questionnaires. The validation of these data for exercise intensity was intricate for distinguishing the borderline between groups 2 and 3. Another concern was the unequal distribution of menopausal women among the groups; groups 1, 2, and 3 contained 14, 5, and 11 postmenopausal women, respectively. These unequal distributions of menopausal status between groups might interfere with the interpretation of the current results. Because menopausal women displayed more atherogenic lipid and lipoprotein profiles with increased dysfunctional HDL [36], this might explain why group 2 showed more LDL aggregation in the loading position (lane 2, Figure 2A) and smaller HDL_2_ particle size than group 3. In a future study, more details on the in vivo functionality and anti-inflammatory properties of HDL from each participant should be investigated to observe trends in different parameters across participants. It would be useful information if we knew which parameters are more influential on HDL quality and functionality in vivo across participants. 

A strength of this study is that the current report is the first report to show that HDL quality could be improved by regular and habitual exercise in middle-aged women (around their fifties) who have a declined metabolic rate and reduced physical activity. It has been reported that HDL-C in women sharply decreased between the ages of 40 and 60 years [37], and the long-term consumption of ethanol in middle-aged women induced the impairment of HDL quality, making it atherogenic [10]. Therefore, it is likely that middle-aged women who drink occasionally and are sedentary would have lower HDL-C and impaired HDL quality and functionality. However, the current study clearly showed that HDL quantity and quality can be remarkably enhanced by regular exercise in an intensity-dependent manner.

In conclusion, this study examined middle-aged women who were sedentary versus those who regularly exercised in their daily lives. The results show that higher exercise intensity is associated with an improved lipid profile, lower LDL-C, higher HDL-C, the amelioration of hepatic functions, and lower AST levels compared to those observed in participants with a sedentary lifestyle. In addition, the lipoprotein and apolipoprotein profiles were improved in the exercise groups (groups 2 and 3) compared to the sedentary group. The LDL properties and cholesterol levels were augmented, and there was less oxidation in the exercise group than in the sedentary lifestyle group. HDL quality and functionality were enhanced, exhibiting a larger particle size, a distinct particle shape, enriched cholesterol content, and elevated antioxidant activities.

## 4. Materials and Methods

### 4.1. Participants

This study was initiated to identify the characteristics of HDL and LDL regarding quantity and quality among a healthy Korean population between 20 and 70 years old through a nationwide advertisement in Korea from 2021 to 2022. Female middle-aged (35–63 years old) volunteers were recruited randomly, and we found that they could be divided into sedentary and exercise groups depending on metabolic equivalents (METs) per week. After recruiting middle-aged women volunteers for the retrospective study, they were divided into three groups: group 1 (sedentary, MET 2.5 ± 0.03), group 2 (low-intensity exercise, MET 4.5 ± 0.3), and group 3 (high-intensity exercise, MET 9.1 ± 0.4)

This study was approved by the Korea National Institute for Bioethics Policy (KoNIBP, approval number P01-202109-31-009) supported by the Ministry of Health Care and Welfare (MOHW) of Korea. All participants were of Korean ethnic origin, and they consumed a typical Korean diet, which is enriched with rice-based carbohydrates (60.8%), total fats (24.2%), and proteins (15.1%), consisting of vegetables, meat, and fish. There were no vegan or kosher diets among the participants.

The exercise frequency, duration, and intensity were estimated from a self-administered questionnaire inquiring about the frequency, time, and intensity of exercise per week during the 1 year prior. Therefore, we assume that participants in groups 2 and 3 had been exercising for at least 1 year or more. METs were calculated using a shorthand method for estimating energy expenditure during physical activity. In accordance with the previous study, MET assigns the intensity values of specific activities [23]. The MET scores were calculated based on the survey results of the participants and were classified as follows: Light < 3.0 METs; Moderate 3.0~6.0 METs; Vigorous > 6.0 METs. 

### 4.2. Anthropometric Analysis

Blood pressure was measured using an Omron HBP-9020 (Kyoto, Japan). Height, body weight, body mass index (BMI), body water, total body fat (%), total body fat mass (kg), and visceral fat mass (VFM) (kg) were measured individually using an X-scan plus II body composition analyzer (Jawon Medical, Gyeongsan, Republic of Korea). Handgrip strength (HS) was measured in the standing position with the arms straight down to the sides. The maximum grip strengths of the right and left hands were measured three times alternatively using a digital hand dynamometer (digital grip strength dynamometer, T.K.K 5401; Takei Scientific Instruments Co., Ltd., Tokyo, Japan). After the handgrip strengths of both hands were measured, a 60 s rest interval was allowed. The maximum grip strength of the dominant hand was used for the analysis [38].

### 4.3. Blood Analysis

After fasting overnight, blood was collected using a vacutainer (BD Bio Sciences, Franklin Lakes, NJ, USA) without adding an anticoagulant. The serum parameters in Table 2 were determined with an automatic analyzer using CobasC502 (Roche, Germany) at a commercial diagnostic service via SCL healthcare (Seoul, Republic of Korea).

### 4.4. Isolation of Lipoproteins

Very low-density lipoprotein (VLDL, d < 1.019 g/mL), LDL (1.019 < d < 1.063), HDL_2_ (1.063 < d < 1.125), and HDL_3_ (1.125 < d < 1.225) were isolated from individual patient and control sera via sequential ultracentrifugation [14,15], with the density adjusted by adding NaCl and NaBr in accordance with standard protocols [39]. The samples were centrifuged for 24 h at 10 °C and 100,000× g using a Himac NX (Hitachi, Tokyo, Japan) at the Raydel Research Institute (Daegu, Republic of Korea).

For each of the lipoproteins purified individually, the total cholesterol (TC) and TG measurements were obtained using commercially available kits (cholesterol, T-CHO, and TG, Cleantech TS-S; Wako Pure Chemical, Osaka, Japan). The protein concentrations of the lipoproteins were determined using a Lowry protein assay, as modified by Markwell et al. [40], using the Bradford assay reagent (Bio-Rad, Seoul, Republic of Korea) with bovine serum albumin (BSA) as a standard.

### 4.5. LDL Oxidation and Quantification

The degree of oxidation of individual LDL was assessed by measuring the concentration of oxidized species in LDL according to the thiobarbituric acid-reactive substances (TBARS) method using malondialdehyde (MDA) as a standard [41].

Oxidized LDL (oxLDL) was produced by incubating the LDL fraction with CuSO_4_ (final concentration, 10 μM) for 4 h at 37 °C. oxLDL was then filtered (0.2 μm) and analyzed using a thiobarbituric acid-reactive substances (TBARS) assay to determine the extent of oxidation, as described elsewhere [41].

### 4.6. Paraoxonase Assay

The paraoxonase-1 (PON-1) activity in HDL_2_ and HDL_3_ toward paraoxon was determined by evaluating the hydrolysis of paraoxon into *p*-nitrophenol and diethylphosphate, which is catalyzed by the enzyme [42]. The PON-1 activity was then determined by measuring the initial velocity of *p*-nitrophenol production at 37 °C, as determined by measuring the absorbance at 415 nm (microplate reader, Bio-Rad model 680; Bio-Rad, Hercules, CA, USA), as described previously [42,43].

### 4.7. Ferric-Ion-Reducing Ability Assay

The ferric-ion-reducing ability (FRA) was determined using the method reported by Benzie and Strain [44]. Briefly, the FRA reagent was freshly prepared by mixing 20 mL of 0.2 M acetate buffer (pH 3.6), 2.5 mL of 10 mM 2,4,6-tripyridyl-S-triazine (Fluka Chemicals, Buchs, Switzerland), and 2.5 mL of 20 mM FeCl_3_∙6H_2_O. The antioxidant activities of HDL (2 mg/mL) were estimated by measuring the increase in absorbance induced by the ferrous ions generated. The freshly prepared FRA reagent (300 μL) was mixed with OSO and SO as an antioxidant source. FRA was determined by measuring the absorbance at 593 nm every 2 min over a 60 min period at 25 °C using a UV-2600i spectrophotometer.

### 4.8. Electromobility of Lipoproteins

The electromobility of the participants’ samples was compared by evaluating the migration of each lipoprotein (LDL, HDL_2_, and HDL_3_) by agarose electrophoresis [45]. The relative electrophoretic mobility depends on the intact charge and three-dimensional structure of HDL. Therefore, for each group, agarose gel electrophoresis with HDL_2_ and HDL_3_ was carried out in the nondenatured state [46]. The gels were then dried and stained with 0.125% Coomassie Brilliant Blue, after which the relative band intensities were compared by band scanning using Gel Doc^®^ XR (Bio-Rad) with Quantity One software (version 4.5.2).

### 4.9. Glycation of HDL

Purified HDL (2 mg/mL) was incubated with 250 mM D-fructose in 200 mM potassium phosphate/0.02% sodium azide buffer (pH 7.4) for up to 72 h in air containing 5% CO_2_ at 37 °C. Fructose can induce a remarkably greater extent of apoA-I glycation than glucose, according to a previous report [47]. The extent of glycation was determined by measuring the fluorometric intensity at 370 nm (excitation) and 440 nm (emission), as described previously [48], using an LS55 spectrofluorometer (PerkinElmer) and a 1 cm path-length suprasil quartz cuvette (Fisher Scientific, Pittsburg, PA, USA).

### 4.10. Electron Microscopy

Transmission electron microscopy (TEM, Hitachi H-7800; Ibaraki, Japan) was performed at the Raydel Research Institute (Daegu, Republic of Korea) at an acceleration voltage of 80 kV. HDL was negatively stained with 1% sodium phosphotungstate (PTA; pH 7.4) with a final apolipoprotein concentration of 0.3 mg/mL in TBS. A volume of 5 μL of the HDL suspension was blotted with filter paper and replaced immediately with a 5 μL droplet of 1% PTA. After a few seconds, the stained HDL fraction was blotted onto a Formvar carbon-coated 300 mesh copper grid and air-dried. The shape and size of the HDL particles were determined by TEM at 40,000× magnification, according to a previous report [43,46].

### 4.11. Data Analysis

All analyses in Table 1, Table 2, Table 3 and Table 4 were normalized using a homogeneity test of variances through Levene’s statistics. Nonparametric statistics were performed using a Kruskal–Wallis test if not normalized. All values are expressed as the median (25th; 75th percentiles) in Table 1 and Table 2 for continuous variables for the middle-aged women groups. Data in Table 3 are expressed as the mean ± SEM (standard error of the mean). Multiple groups were compared using a one-way analysis of variance (ANOVA), and the results are reported in Table 1, Table 2 and Table 3. All tests were two-tailed, and the statistical significance was defined at *p* < 0.05.

The anthropometric profiles of the middle-aged women groups depend on exercise time and intensity (Table 1). Anthropometric profiles, such as alcohol intake amount, height, weight, heart rate, SBP, DBP, muscle mass, fat mass, and body water, were compared using ANOVA. Age, visceral fat mass, HS, and BMI were compared using a Kruskal–Wallis test.

The blood lipid and inflammatory parameters of the middle-aged women groups depend on the exercise time and intensity (Table 2). Blood lipid and inflammatory parameters, such as HDL-C, HDL-C/TC, LDL-C, LDL-C/HDL-C, apo-B, glucose, hs-CRP, and AST, were compared using ANOVA. TC, TG, TG/HDL-C, apoA-I, apo-B/apoA-I, ALT, and γ-GTP were compared using a Kruskal–Wallis test.

The characteristics of lipoproteins from the middle-aged women groups depend on the exercise time and intensity (Table 3). The characteristics of the lipoproteins, such as LDL-MDA, LDL-TG, HDL_2_-PON, HDL_3_-TC, and HDL_3_-PON, were compared using ANOVA. The LDL size, LDL glycation, LDL-TC, HDL_2_ size, HDL_2_ glycation, HDL_2_-TC, HDL_2_-TG, HDL_3_ size, and HDL_3_-TG were compared using a Kruskal–Wallis test.

A Bonferroni test was used as a post hoc analysis to determine the significance of the differences in continuous variables to identify differences between groups. Spearman correlation analysis was conducted to find a positive or negative association (Table 4). Statistical analyses were carried out using the SPSS statistical package version 28.0 (SPSS Inc., Chicago, IL, USA), incorporating sampling weights and adjusting for the complex survey design.

## Figures and Tables

**Figure 1 ijms-24-01151-f001:**
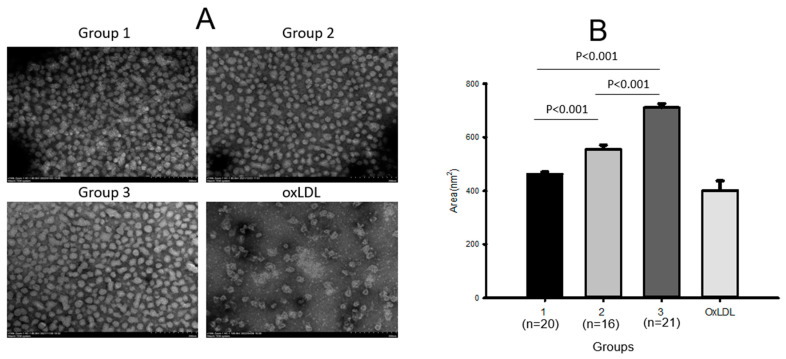
Transmission electron microscopy (TEM) images (**A**) and area analysis (**B**) of LDL from each group.

**Figure 2 ijms-24-01151-f002:**
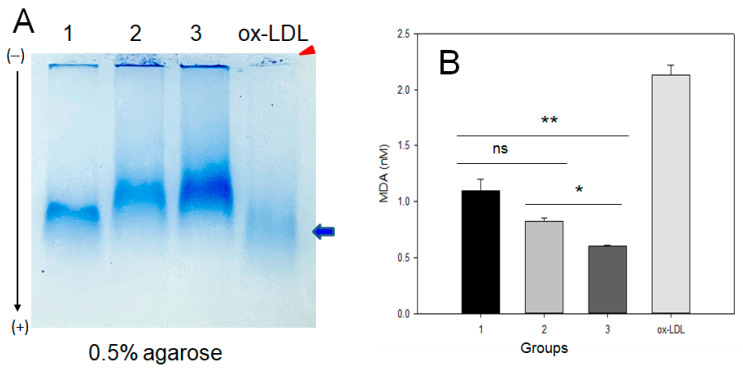
Native electrophoresis of LDL on agarose from each group and quantification of oxidized species. (**A**) Comparison of electromobility and aggregation extent of LDL (10 μg) band on 0.5% agarose gel without denaturation. Lane 1, group 1 (n = 20); lane 2, group 2 (n = 16); lane 3, group 3 (n = 21); lane 4, oxLDL, Cu^2+^-treated for 4 h. Red arrowhead indicates an aggregated band of oxLDL at the loading position. The blue arrow indicates the oxLDL band position and smeared intensity. (**B**) Quantification of oxidized species using malondialdehyde standard by the thiobarbituric acid-reactive substances (TBARS) assay. MDA, malondialdehyde; oxLDL, oxidized LDL. *, *p* < 0.05 between group 2 and 3; **, *p* < 0.01 between group 1 and 3.

**Figure 3 ijms-24-01151-f003:**
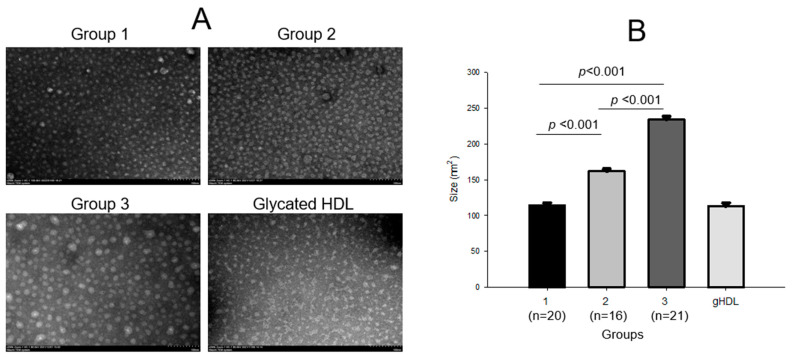
Transmission electron microscopy (TEM) images and area analysis of HDL_2_ from each group (**A**) and size comparison between the groups (**B**). gHDL, glycated HDL.

**Figure 4 ijms-24-01151-f004:**
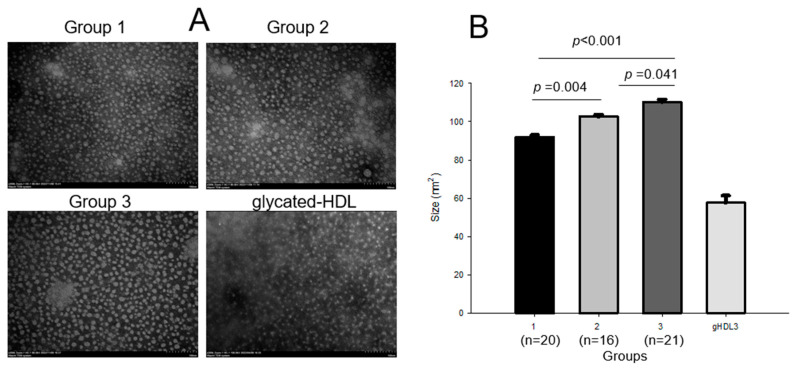
Transmission electron microscopy (TEM) images and area analysis of HDL_3_ from each group (**A**) and comparison of particle sizes (**B**). gHDL_3_, glycated HDL_3_.

**Figure 5 ijms-24-01151-f005:**
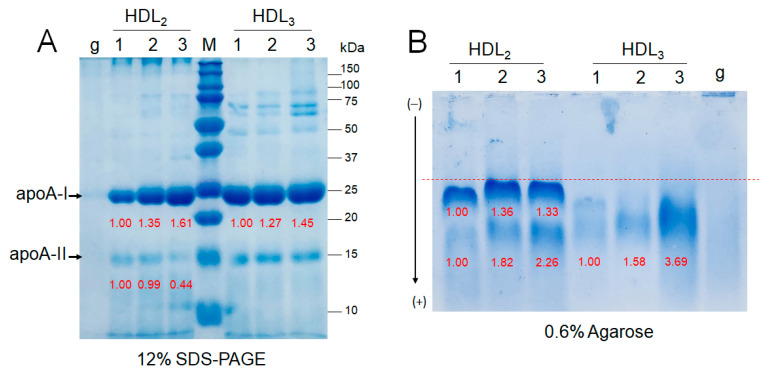
Electrophoretic patterns of HDL from participants depend on the exercise intensity. Lane 1, group 1 (n = 20); lane 2, group 2 (n = 16); lane 3, group 3 (n = 21). (**A**) Electrophoresis of HDL (2 mg/mL) under denatured state (12% SDS-PAGE). Lane M, molecular weight standards (Bio-Rad Cat#161-0374); lane g, glycated HDL_2_. (**B**) Electrophoresis of HDL (2 mg/mL) under native state (0.6% agarose). Lane g, glycated HDL_3_**.**

**Figure 6 ijms-24-01151-f006:**
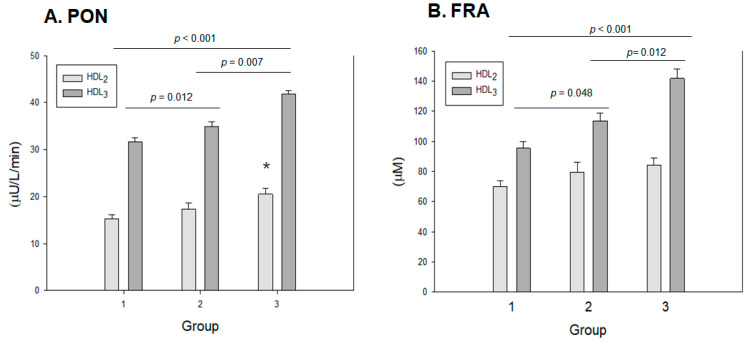
Antioxidant abilities of HDL_2_ and HDL_3_ from each group. Group 1 (n = 20); group 2 (n = 16); group 3 (n = 21). (**A**) HDL-associated paraoxonase (PON) activity. PON activity is expressed as the initial velocity of *p*-nitrophenol production per min (μU/L/min) at 37 °C during 60 min incubation. *, *p* < 0.05 versus group 1. (**B**) Comparison of HDL-associated ferric ion reduction ability (FRA). FRA is expressed as the concentration of vitamin C (mM), which is equivalent to the amount of ferric ions (μM) reduced per hour.

**Table 2 ijms-24-01151-t002:** Blood lipid and inflammatory parameters of women groups depending on exercise (Me (25%; 75%)) *.

	Group 1 n = 20	Group 2 n = 16	Group 3 n = 21	*p*	1 vs. 2	1 vs. 3	2 vs. 3
MET equivalents of common physical activities (score)	2.5 (2.5; 2.5)	4.8 (3.3; 5.5)	9.5 (8.0; 10.0)	0.000	0.000	0.000	0.000
TC (mg/dL)	222.0 (195.2; 244.1)	210.8 (170.3; 251.0)	206.0 (184.3; 220.8)	0.179	1.000	0.199	0.924
HDL-C (mg/dL)	46.7 (42.3; 56.5)	52.5 (45.4; 59.7)	54.0 (47.8; 61.9)	0.030	0.133	0.040	1.000
HDL-C/TC (ratio)	0.22 (0.19; 0.24)	0.26 (0.20; 0.34)	0.28 (0.24; 0.30)	0.002	0.099	<0.001	0.921
LDL-C (mg/dL)	151.5 (131.8; 170.3)	141.0 (100.5; 178.8)	137.0 (115.5; 141.5)	0.172	0.764	0.104	0.850
TG (mg/dL)	107.5 (85.4; 136.6)	78.0 (57.5; 114.9)	57.2 (44.5; 65.5)	<0.001	0.165	<0.001	0.080
TG/HDL-C (ratio)	2.4 (1.5; 3.3)	1.5 (1.0; 2.5)	1.0 (0.9; 1.4)	<0.001	0.183	<0.001	0.112
LDL-C/HDL-C (ratio)	3.1 (2.8; 3.7)	2.3 (1.8; 3.7)	2.4 (2.2; 3.0)	0.004	0.277	0.001	0.652
ApoA-I (mg/dL)	163.0 (153.0; 180.5)	167.5 (152.5; 210.8)	183.0 (161.5; 200.0)	0.072	0.223	0.101	1.000
ApoB (mg/dL)	103.5 (91.5; 117.8)	104.5 (73.8; 118.3)	91.0 (80.5; 100.0)	0.132	0.919	0.136	1.000
ApoB/ApoA-I (ratio)	0.64 (0.50; 0.82)	0.55 (0.41; 0.70)	0.54 (0.44; 0.61)	0.014	0.261	0.009	0.797
RC (mg/dL)	21.1 (16.9;27.2)	15.6 (11.3;22.6)	11.4 (8.9;13.0)	<0.001	0.166	<0.001	0.097
Glucose (mg/dL)	96.0 (90.0; 101.8)	89.0 (82.5; 93.0)	84.0 (80.5; 92.0)	0.003	0.028	0.003	1.000
hs-CRP (mg/L)	0.24 (0.15; 0.55)	0.28 (0.23; 0.48)	0.32 (0.21; 0.51)	0.896	1.000	1.000	1.000
AST (Unit/L)	20.0 (17.0; 22.8)	16.5 (11.5; 18.0)	14.0 (12.5; 19.0)	0.003	0.028	0.003	1.000
ALT (Unit/L)	14.0 (11.0; 16.0)	11.0 (7.3; 16.8)	11.0 (8.0; 14.0)	0.088	0.314	0.110	1.000
γ-GTP (Unit/L)	14.5 (10.3; 25.0)	12.5 (9.3; 18.0)	10.0 (8.0; 15.0)	0.064	0.943	0.058	0.708

ApoA-I, apolipoprotein A-I; Apo-B, apolipoprotein B; AST, aspartate aminotransferase; ALT, alanine transaminase; HDL-C, high-density lipoprotein-cholesterol; hs-CRP, high-sensitivity C-reactive protein; LDL-C, low-density lipoprotein-cholesterol; γ-GTP, gamma-glutamyl transferase; SEM, standard error of the mean; RC, remnant cholesterol; TC, total cholesterol; TG, triglyceride. * Data are presented as median (25th; 75th percentiles). The metabolic equivalent (MET) score was calculated based on the survey results of the participants and can be classified as previously described [23]: Light < 3.0 METs; Moderate 3.0~6.0 METs; Vigorous > 6.0 METs.

**Table 3 ijms-24-01151-t003:** Lipid compositions, oxidation extent, and glycation extent of lipoproteins from participants.

		Group 1 Sedentary	Group 2 Low- Exercise	Group 3 High- Exercise	*p*	1 vs. 2	1 vs. 3	2 vs. 3
MET score/week		2.48 ± 0.03	4.55 ± 0.27	9.13 ± 0.35	<0.001	<0.001	<0.001	<0.001
LDL	TC (mg/mL)	0.99 ± 0.17	1.09 ± 0.09	1.12 ± 0.10	0.801	1.000	1.000	1.000
	TG (mg/mL)	0.22 ± 0.03	0.21 ± 0.02	0.17 ± 0.01	0.023	0.095	0.041	0.192
	MDA (nM)	1.78 ± 0.06	1.43 ± 0.11	1.07 ± 0.14	<0.001	0.095	<0.001	0.073
	FI (glycated)	1406 ± 37	1403 ± 78	1317 ± 70	0.572	1.000	1.000	1.000
	Diameter (nm)	23.9 ± 0.4	24.6 ± 0.3	27.9 ± 0.5	<0.001	<0.001	<0.001	<0.001
HDL_2_	TC (mg/mL)	0.55 ± 0.07	0.71 ± 0.13	0.85 ± 0.07	0.031	0.166	0.042	1.000
	TG (mg/mL)	0.14 ± 0.01	0.13 ± 0.02	0.10 ± 0.02	0.541	0.860	0.125	1.000
	MDA (nM)	0.40 ± 0.10	0.31 ± 0.07	0.36 ± 0.09	0.775	1.000	1.000	1.000
	FI (glycated)	934 ± 92.8	853 ± 35	843 ± 60	0.592	1.000	1.000	1.000
	Diameter (nm)	12.6 ± 0.3	15.0 ± 0.3	17.6 ± 0.3	<0.001	<0.001	<0.001	<0.001
HDL_3_	TC (mg/mL)	0.37 ± 0.03	0.36 ± 0.02	0.36 ± 0.02	0.881	0.943	0.988	0.987
	TG (mg/mL)	0.09 ± 0.01	0.06 ± 0.01	0.05 ± 0.01	0.022	0.051	0.017	0.873
	MDA (nM)	0.29 ± 0.05	0.25 ± 0.06	0.24 ± 0.04	0.138	0.074	0.145	0.150
	FI (glycated)	642 ± 46	663 ± 87	713 ± 29	0.473	1.000	0.699	1.000
	Diameter (nm)	10.9 ± 0.2	12.3 ± 0.2	14.6 ± 0.3	<0.001	<0.001	<0.001	<0.001

MET, metabolic equivalent; FI, fluorescence intensity (Ex = 370 nm, Em = 440 nm, 0.01 mg/mL); FRA, ferric ion reduction ability; MDA, malondialdehyde; TC, total cholesterol; TG, triglyceride. Data are expressed as mean and SEM. The metabolic equivalent (MET) score was calculated based on the survey results of the participants and can be classified as previously described [23]: Light < 3.0 METs; Moderate 3.0~6.0 METs; Vigorous > 6.0 METs.

**Table 4 ijms-24-01151-t004:** Spearman correlation (r) and *p* values of linear regression for all participants (n = 57).

		r	*p*
Anthropometric profiles	BMI (kg/m^2^)	−0.072	0.592
	Fat mass_subcutaneous (kg)	−0.166	0.217
	Fat mass_visceral (kg)	−0.167	0.216
	Heart rate (BPM)	0.344	0.009
	SBP (mmHg)	−0.075	0.577
	DBP (mmHg)	−0.178	0.186
	HS (kg)	0.126	0.351
Blood lipid and inflammatory parameters	HDL-C (mg/dL)	0.365	0.005
	ApoA-I (mg/dL)	0.354	0.007
	ApoB (mg/dL)	−0.264	0.047
	ApoB/ApoA-I (ratio)	−0.373	0.004
	LDL-C (mg/dL)	−0.246	0.065
	LDL-C/HDL-C (ratio)	−0.437	0.001
	Glucose	−0.522	0.000
	TC (mg/dL)	−0.234	0.080
	HDL-C/TC (ratio)	0.484	0.000
	TG (mg/dL)	−0.577	0.000
	TG/HDL-C (ratio)	−0.612	0.000
	Remnant cholesterol	−0.572	0.000
	AST (Unit/L)	−0.555	0.000
	ALT (Unit/L)	−0.351	0.007
	γ-GTP (Unit/L)	−0.365	0.005
	hs-CRP (mg/L)	0.116	0.389
Lipoprotein profiles and characteristics	LDL-TC (mg/mL/protein)	0.151	0.491
	LDL-TG (mg/mL/protein)	0.263	0.225
	LDL_MDA (nM)	−0.238	0.000
	LDL_size (nm)	0.272	0.000
	HDL_2_-TC (mg/mL/protein)	0.621	0.002
	HDL_2_-TG (mg/mL/protein)	0.240	0.270
	HDL_2__MDA (nM)	−0.005	0.981
	HDL_2__size (nm)	0.650	0.000
	HDL_2__FRA	0.308	0.153
	HDL_2__PON	0.446	0.033
	HDL_3_-TC (mg/mL/protein)	−0.084	0.702
	HDL_3_-TG (mg/mL/protein)	−0.172	0.433
	HDL_3_-MDA (nM)	0.370	0.082
	HDL_3_-size (nm)	0.485	0.000
	HDL_3_-FRA	0.578	0.004
	HDL_3_-PON	0.811	0.000

ALT, alanine transaminase; AST, aspartate aminotransferase; ApoA-I, apolipoprotein A-I; Apo-B, apolipoprotein B; BMI, body mass index; DBP, diastolic blood pressure; γ-GTP, gamma-glutamyl transferase; HDL, high-density lipoprotein; HDL-C, high-density lipoprotein-cholesterol; HS, handgrip strength; hs-CRP, high-sensitivity C-reactive protein; LDL, low-density lipoprotein; LDL-C, low-density lipoprotein-cholesterol; MDA, *malondialdehyde*; PON, paraoxonase; SBP, systolic blood pressure; TC, total cholesterol; TG, triglyceride.

## Data Availability

The data used to support the findings of this study are available from the corresponding author upon reasonable request.

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
