# Peer review of "Enhancement of High-Density Lipoprotein (HDL) Quantity and Quality by Regular and Habitual Exercise in Middle-Aged Women with Improvements in Lipid and Apolipoprotein Profiles: Larger Particle Size and Higher Antioxidant Ability of HDL"

_ijms, 2023, doi:10.3390/ijms24021151_

Round 1
Reviewer 1 Report
This paper investigates the role of exercise in lipid profiles in middle age women of normal BMI. Consistent with reports in the literature, exercise induced positive changes in lipid profiles, favoring increases in HDL and decreases in LDL. This again underscores the importance of exercise in maintaining health. I have a few questions for consideration in the discussion as detailed below.
1) Do you think these findings would extrapolate to women with higher BMIs? All the women in the study had a healthy BMI.
2) Diet is mentioned as a factor, but there is no data about the diet of the patient population. Had you considered collecting this information?
3) How do you think diet and geographic location might affect these results? Would they extend to Western populations? The Western diet is vastly different from the average diet in Korea.
4) Menopause is clearly a factor, as mentioned in the discussion. Did you consider a multivariate analysis controlling for menopause as a variable?
5) The study population is relatively small with 20 or less participants per group. Do you think there are other significant differences that are not being detected because of power?
I also have a few other points:
1) It would be nice to see each patient graphed as a separate data point, so we could observe trends in different parameters across patients.
2) It is not clear what the error bars represent on the graphs.
3) N values should be listed in the legends.
4) I would refrain from using the word “ordinary” to describe the patient population. Please simply remove this word. Middle age women is sufficient. We do not know if they are truly ordinary.
Author Response
Thank you very much heartily for reviewing and critical comments to improve this paper.
Please find attached pdf as response

Reviewer 2 Report
In the current manuscript, the authors have described the beneficial role of long term exercise in middle aged women. The manuscript described an increase in HDL-C and apoA-I with an enhancement of HDL quality and functionality. The manuscript is well written. The authors need to address the following concerns.
1. The reason behind the selection of middle aged women is not written clearly.
2. The significance of the study is need to be explained in better precise way in the discussion section.
Author Response
Thank you very much heartily for reviewing and critical comments to improve this paper.
Please find attached doc as response and reflections
